# Effects of (2R,6R)-hydroxynorketamine in assays of acute pain-stimulated and pain-depressed behaviors in mice

**Todd M. Hillhouse**[1☯], **Kaitlyn J. Partridge**[1,2☯], **Patrick I. Garrett**[3], **Sarah C. Honeycutt**[4], **Joseph H. Porter**[5,6]*

**1** Department of Psychology, University of Wisconsin Green Bay, Green Bay, Wisconsin, United States of America, **2** Department of Cell Biology, Neurobiology, and Anatomy, Medical College of Wisconsin, Milwaukee, Wisconsin, United States of America, **3** Department of Pharmacology and Toxicology, Brody School of Medicine, East Carolina University, Greenville, North Carolina, United States of America, **4** Department of Psychology, University at Buffalo, Buffalo, New York, United States of America, **5** Department of Psychology, Virginia Commonwealth University, Richmond, Virginia, United States of America, **6** Department of Psychological Sciences, Northern Michigan, Marquette, Michigan, United States of America

☯ These authors contributed equally to this work.

* jporter@vcu.edu

**Data Availability Statement:** Hillhouse, Todd (2024). Effects of (2R,6R) HNK on pain stimulated

## Abstract

Ketamine has been shown to produce analgesia in various acute and chronic pain states; however, abuse liability concerns have limited its utility. The ketamine metabolite (2R,6R)-hydroxynorketamine (HNK) has been shown to produce antidepressant-like effects similar to ketamine without abuse liability concerns. (2R,6R)-HNK produces sustained analgesia in models of chronic pain, but has yet to be evaluated in models of acute pain. The present study evaluated the efficacy of acute (2R,6R)-HNK administration (one injection) in assays of pain-stimulated (52- and 56-degree hot plate test and acetic acid writhing) and pain-depressed behavior (locomotor activity and rearing) in male and female C57BL/6 mice. In assays of pain-stimulated behaviors, (2R,6R)-HNK (1–32 mg/kg) failed to produce antinociception in the 52- and 56-degree hot plate and acetic acid writhing assays. In assays of pain-depressed behaviors, 0.56% acetic acid produced a robust depression of locomotor activity and rearing that was not blocked by pretreatment of (2R,6R)-HNK (3.2–32 mg/kg). The positive controls morphine (hot plate test) and ketoprofen (acetic acid writhing, locomotor activity, and rearing) blocked pain-stimulated and pain-depressed behaviors. Finally, the effects of intermittent (2R,6R)-HNK administration were evaluated in 52-degree hot plate and pain-depressed locomotor activity and rearing. Intermittent administration of (2R,6R)-HNK also did not produce antinociceptive effects in the hot plate or pain-depressed locomotor activity assays. These results suggest that (2R,6R)-HNK is unlikely to have efficacy in treating acute pain; however, the efficacy of (2R,6R)-HNK in chronic pain states should continue to be evaluated.

and pain depressed behaviors. figshare. Dataset.
https://doi.org/10.6084/m9.figshare.25289362.v1.

**Funding:** The author(s) received no specific funding for this work.

**Competing interests:** The authors have declared that no competing interests exist.

## Introduction

Inadequate pain relief from traditional pain medication and the ongoing opioid crisis have created a crucial need for novel analgesics [1,2]. The dissociative anesthetic, ketamine, has been shown to be effective in treating acute, chronic, and cancer-related neuropathic pain in humans [3,4]. However, the route of ketamine administration (intravenous) and unwanted side effects (e.g. dissociative hallucinogenic effects, sedation, cognitive impairments, abuse liability, liver injury, etc.) have limited the utility of ketamine to a general patient population. The ketamine metabolite, (2R,6R)-HNK, has emerged as a promising candidate for the treatment of acute and chronic pain for several reasons; (2R,6R)-HNK exhibits significant plasma and brain concentrations following administration (oral or intraperitoneal), appears to be void of psychotomimetic and abuse-related effects, and produces antidepressant-like effects when administered alone [5–8]. Acute and three days of repeated treatment with (2R,6R)-HNK also has been shown to reverse chronic pain in mouse models which include complex regional pain syndrome type-1, plantar incision postoperative pain, spared nerve injury, formalin test, low-frequency percutaneous electrical nerve stimulation (LF-PENS), and mechanical hypersensitivity produced by carrageenan in the hind paw [9–13]. The present study sought to extend the findings on the antinociceptive effects of (2R,6R)-HNK in assays of acute pain.

To date, (2R,6R)-HNK has only been evaluated in assays of pain-stimulated behaviors, which exclusively rely on behaviors that increase in rate, intensity, and frequency following the presentation or administration of a noxious stimulus [14–16]. Pain-stimulated behaviors can be attenuated/reversed by drugs that block or reduce the sensitivity to the noxious stimulus *OR* by drugs that produce motor sedation or impair general locomotor activity (e.g. haloperidol or kappa agonists). Although results from pain-stimulated studies are promising and can provide valuable information about candidate analgesics, complete reliance on assays of pain-stimulated behavior may prove to be problematic leading to false positives (impaired motor function as described above). Assays of pain-depressed behaviors can be defined as behaviors that decrease in rate, intensity, or frequency following to administration/presentation of a noxious stimulus and used alone or to complement assays of pain-stimulated behavior [15–17]. These pain-depressed behavioral assays provide a translation aspect that is missing with pain-stimulated behavior, as people dealing with pain typically decrease their activity and work productivity [18]. Additionally, pain-depressed assays are able to identify compounds that produce false positives of pain-stimulated due to motor impairment as these compounds would not be able increase behaviors back to baseline levels. In the present study, we used an assay of pain-depressed locomotor activity as both acetic and lactic acid produce a robust depression of locomotor activity that can be reversed with nonsteroidal anti-inflammatory drug (NSAID) as well as standard opioid drugs [17,19]. In addition to determining if a drug can reverse the acid-induced decrease in locomotor activity and rearing, this assay allowed us to evaluate the effects of each drug on locomotor activity in the absence of acid. Specifically, the combination of drug + DH2O (drug alone) is used to rule out false positives as an increase in locomotor activity in the absence and presence of the acetic acid indicated a nonselective increase in locomotor activity—not an antinociceptive effect, Additionally, a decrease in locomotor activity in the absence and presence of the acetic acid is a nonselective suppression of behavior resulting in a false negative.

Here, we evaluated the antinociceptive effects of (2R,6R)-HNK in assays of both acute pain-stimulated and pain-depressed behaviors in male and female mice. Acute pain-stimulated assays included 52°C hot plate, 56°C hot plate, and acid-stimulated writhing. Additionally, the ability of (2R,6R)-HNK to restore pain-depressed locomotor activity and rearing in mice was assessed. Finally, the ability of intermittent (2R,6R)-HNK administration to block pain-

stimulated withdrawal latency in the 52°C hot plate and pain-depressed locomotor activity and rearing were evaluated. The antinociceptive effects of (2R,6R)-HNK were compared to the positive controls, morphine and ketoprofen.

## Materials and methods

### Animals

Adult (>8 weeks of age) male and female C57BL/6 mice that were bred in-house University of Wisconsin-Green Bay or Weber State University were used for all experiments. C57BL/6 breeder mice were purchased from Envigo (Indianapolis, IN). A temperature (20 ± 3˚C) and light-controlled vivarium housed mice with all experimental testing occurring during the light phase (lights on 0600–1800 hours). Mice were group-housed (n = 2–4 per cage) in standard shoebox cages with small corn cob bedding (Envigo) and given ad libitum food (18% protein Teklad diet) and water prior to experimental testing. All experimental procedures complied with federal guidelines and were approved by the Institutional Animal Care and Use Committee at the University of Wisconsin-Green Bay and Weber State University [20].

### Drugs and chemicals

(2R,6R)-HNK was provided by the National Center for Advancing Translational Sciences (NCATS; Bethesda, Maryland, USA; purity >99.5%, certification of analysis is available in the supplementary materials). Ketoprofen and morphine were purchased from Sigma-Aldrich (Saint Louis, MO). Acetic acid was purchased from Fisher Scientific (Ward Hill, MA). Morphine and (2R,6R)-HNK were dissolved in 0.9% saline. Ketoprofen was dissolved in 95% sterile water, 2.5% ethanol, 2.5% Tween 80. Acetic acid was diluted in deionized water until the desired concentration (0.18, 0.32, and 0.56%) was achieved. All compounds were administered intraperitoneally (i.p.) at a volume of 10 ml/kg. Drug doses and pretreatment times were selected based on published bioavailability and behavioral studies [5,7,8,21–24].

### Effects of acute dosing on hot plate test

Experimental procedures were conducted similarly to those described previously [22,25]. A hot plate analgesia meter (IITC Life Sciences Inc., Woodland Hills, CA) was used to assess the antinociception effects of the compounds. On test day, subjects were moved to the test room and provided a 30 min acclimation period. Mice were placed onto the surface of the hot plate (24 x 24 cm) within a 15 cm high clear acrylic square and latency to first forepaw or hindpaw lick, fluttering of hindpaws, or jumping was measured. Two trained coders, who were blind to treatment conditions, scored hot plate behavior and the average of the two latencies was used for data analysis. Between each mouse, the surface of the hot plate and acrylic sides were cleaned with 50% ethanol to remove odor/scent marks. To assess sex differences in C57BL/6 mice (Male N = 9; Female N = 9), a range of increasing temperatures were used (44–56±0.1˚C) with 15 min intervals between test trials. To prevent tissue damage, a 20 s cutoff time was used for all temperatures.

A similar number of male and female mice were used to assess the antinociceptive effects of (2R, 6R)-HNK at temperatures of 52±0.1˚C and 56±0.1˚C with a cutoff time of 30 sec and 20 sec, respectively. Cumulative dosing of (2R, 6R)-HNK was administered at 30 min intervals using half-log increments (Male N = 7–8; Female N = 6–7). Morphine (32 mg/kg) was used as positive control (Male N = 6, Female N = 6 for both temperatures).

## Effects of acute dosing on acid-stimulated writhing and acid-depressed rearing

Writhing experimental procedures were adapted from published literature [14,25,26]. Mice were moved to the testing room 30 min prior to the start of testing for an acclimation period. Mice received i.p. administration of acetic acid and were immediately placed in empty shoebox cages (18.5cm x 29.5cm x 12.5cm) and video recorded for 40 minutes. Between mice, all surfaces (floor and walls) were cleaned with 50% ethanol to remove odor/scent marks. Videos were scored by trained, blind observers for two behaviors, writhing and rearing. Including rearing as a measure has provided us with a measure of pain-stimulated (writhing) and pain-depressed (rearing) behavior within the same assay allowing us to evaluate all behavioral outcomes to completely understand the antinociceptive effects of (2R,6R)-HNK. Writhing was operationally defined as a contraction of the abdomen followed by a stretching of the hind limbs. Rearing was operationally defined as a raise of both forelimbs off the floor of the cage and extending the head upwards. A within-subjects design was utilized for each study, and acetic acid injections were separated by one week to reduce the development of tolerance to the acid effect [14,27–29]. For the acetic acid concentration curve, male (N = 8) and female (N = 7) mice received different acetic acid (0, 0.18, 0.32, and 0.56%) concentrations using a Latin-square design.

For the drug experiments, (2R,6R)-HNK (0–32 mg/kg) and the positive control ketoprofen (0 or 3.2 mg/kg) were administered 30 mins prior to 0.56% acetic acid administration (which was selected based on the results from the acetic acid concentration curve experiment). Drug doses were randomized using a Latin-square design. There was a minimum of 48 hours between test sessions, and acetic acid administration was separated by a week. For example, on Monday mice were injected with a drug + 0.56% acetic acid, and on Thursday mice were injected with a drug + DH2O. Separate groups of mice were used for (2R,6R)-HNK (Male N = 4; Female N = 6) and ketoprofen (Male N = 8; Female N = 7).

## Effects of acute dosing on acid-depressed locomotor activity and rearing

Pain-depressed locomotor activity experimental procedures were adapted from Stevenson et al. [17]. Distance Traveled (locomotor activity) was measured using a standard open field Plexiglas chamber (29 cm × 29 cm × 20 cm) equipped with three 16-beam IR arrays (Med Associates, Inc., St. Albans, VT). Open field chambers were each enclosed in a cabinet equipped with a house light (on the back wall) and a small fan. Distance traveled (cm) and rearing was measured with Med-Associates Activity Monitor (version 7; Med Associates, Inc.). Mice received one habituation session in which mice were removed from their home cages, weighed, and individually placed in the open field for 60 mins (no injections). Between mice, all surfaces (floor and walls) were cleaned with 50% ethanol to remove odor/scent marks. After the habituation session, an acid concentration-effect curve (0-.56%) was conducted in male (N = 9) and female (N = 8) mice. Mice received an injection of acetic acid, and then were immediately placed in the open field chambers for 30 mins to assess pain-depressed activity. Acetic acid tests were separated by one week to be consistent with the writhing studies and reduce the development of tolerance to the acid effects.

For the drug experiments, (2R,6R)-HNK (0–32 mg/kg) and ketoprofen (0 or 3.2 mg/kg) were administered 30 mins prior to 0.56% acetic acid or vehicle (deionized water) injections. Acid vehicle tests were used to determine if (2R,6R)-HNK or ketoprofen could reverse acetic acid depression of LMA without stimulating locomotor activity. Drug doses were randomized using a Latin-square design. To be consistent with the writhing experiments, there was a a minimum of 48 hours between test sessions, and acetic acid administration was separated by a

week. For example, on Monday mice were injected with a drug + 0.56% acetic acid, and on Thursday mice were injected with a drug + DH2O. Separate groups of mice were used for (2R,6R)-HNK (Male N = 9; Female N = 8) and ketoprofen (Male N = 6; Female N = 6).

### Effects of intermittent (2R,6R)-HNK administration on hot plate latency and acid-depressed LMA

Acute administration (1 injection) of (2R,6R)-HNK failed to produce an antinociception response in assays of pain-stimulated and pain-depressed behaviors; therefore, it was decided to evaluate the antinociceptive effects of intermittent administration of (2R,6R)-HNK to help rule out pretreatment time as the mechanism for the lack of antinociception in the present study. Intermittent (2R,6R)-HNK administration procedures were adapted from Wulf et al. [24]. Specifically, separate groups of mice received either saline or 10 mg/kg (2R,6R)-HNK administered three times at 48 h intervals (injections on days 1, 3, and 5). On day 6 (test day), mice received a final injection of saline or (2R,6R)-HNK 30 mins before the test session (Fig 5A provides a visual timeline of injections and behavioral testing). For the hot plate test, a 52 ±0.1˚C temperature was used. Experiments were conducted as described above. Baseline control tests (Saline + DH2O and Saline + 0.56% acetic acid) were determined prior to the start of intermittent administration of saline or (2R,6R)-HNK.

### Data analysis

For the hot plate test, a two-way mixed factors analysis of variance (ANOVA) was used with "sex" as the between-subject factor and "temperature" or "HNK-dose" as the within-subject factor. For the acid-stimulated writhing and acid-depressed LMA, a two-way mixed factors ANOVA was used with "sex" as the between-subject factor and "acetic acid concentration" or "dose" as the within-subject factor. A paired T-test was used to analyze the effects of ketoprofen on acid-stimulated writhing and rearing. When no significant sex effects were found in the baseline control experiments OR in the (2R,6R)-HNK experiments, we reanalyzed the data using a one-way repeated measures ANOVA without sex as a factor. For transparency purposes, figures and analyses that used sex as a factor are available in the supplemental materials (these figures align with Figs 2 and 3 in the main article). For the intermittent dosing experiments, a two-way mixed factor ANOVA was used with "drug treatment" as a between-subject factor and "acid condition" as the within-subject factor. All significant one-way ANOVAs were followed by Dunnett's post-hoc test and significant two-way ANOVAs were followed by a Tukey multiple comparisons post-hoc tests. The criterion for significance was set at $P \leq 0.05$. GraphPad Prism 7.0 (La Jolla, CA) was used to graph and analyze data. All data are expressed as mean ± standard error of the mean (SEM).

## Results

### Effects of acute dosing on hot plate test

There was temperature-dependent decrease in withdrawal latency (Fig 1A; $F_{(4, 60)} = 195.10$, $p < 0.001$); however, there was no main effect for sex ($F_{(1, 15)} = 0.07$, NS) or significant interaction ($F_{(4, 60)} = 1.80$, NS). The Tukey post hoc test revealed that 50˚C, 53˚C, and 56˚C significantly decreased response latency as compared to 44˚C and 47˚C (and each other) with the greatest decrease occurring at 56˚C in both sexes ($P < 0.01$).

(2R,6R)-HNK failed to significantly alter withdrawal latency on the 52ºC hot plate (Fig 1B): Interaction ($F_{(3, 36)} = 1.18$, NS), main effect of dose ($F_{(3, 36)} = 1.88$, NS), and main effect of sex ($F_{(1, 12)} = 0.09$, NS). For the 56ºC hot plate (Fig 1C), treatment with (2R,6R)-HNK

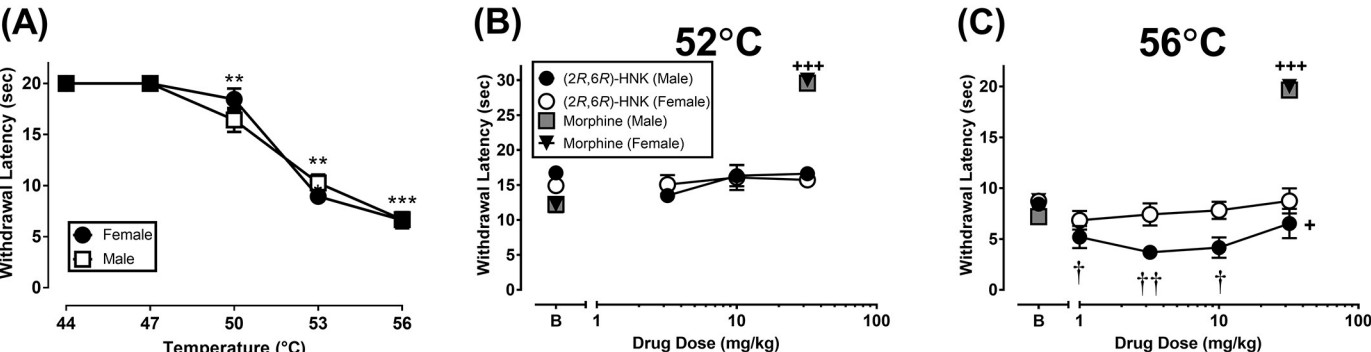

**Fig 1. Effect of sex and acute dosing of (2R,6R)-HNK on withdrawal latency in the hot plate assay.** (A) There was a temperature-dependent decrease in withdrawal latency for male and female mice. The Tukey post hoc test revealed that 50˚C, 53˚C, and 56˚C significantly decreased response latency as compared to 44˚C and 47˚C (and each other) with the greatest decrease occurring at 56˚C in both sexes (P < 0.01). (B) (2R,6R)-HNK did not change withdrawal latencies in the 52ºC hot plate. (C) (2R,6R)-HNK significantly decreased withdrawal latencies in the 56ºC hot plate. The Y-axis highest value represents the cutoff time. All drugs were 30 mins prior to the start of the experiment. All significant ANOVAs were followed by a Tukey post hoc test. **P < 0.01, ***P < 0.001 represents the significant main effect of temperature (temperature dependent effect). †P < 0.05, ††p < 0.01, represent the significant main effect of dose versus baseline. +++P < 0.001 represents significant increase as compared to morphine baseline All data show mean ± SEM. Panel A (Male N = 9, Female N = 9); Panel B (2R,6R)-HNK (Male N = 8, Female N = 6); Panel C (2R,6R)-HNK (Male N = 7, Female N = 7); Morphine (Male N = 6, Female N = 6 for both temperatures).

produced a significant main effect of dose (F(4, 48) = 5.42, P = 0.001) and sex (F(1, 12) = 5.64, P < 0.05); however, the interaction was not significant (F(4, 48) = 1.69, NS). The Tukey post hoc test revealed that 1, 3.2, and 10 mg/kg significantly decreased withdrawal latencies as compared to baseline latencies (P < 0.05). Additionally, male mice had significantly lower withdrawal latencies as compared to female mice (P < 0.05). The positive control morphine significantly increased withdrawal latencies to the maximum time in both male and female mice for both 52ºC (t(11) = 25.26, P < 0.001) and 56ºC (t(11) = 27.33, P < 0.001) hot plate (Fig 1B and 1C, respectively).

The 30 min pretreatment time failed to produce any indication of an antinociceptive response by (2R,6R)-HNK. Thus, the same mice were tested again in the 56ºC hot plate 24 hours after the final cumulative dose to align with the pretreatment time in the Yost et al. (2022b) study. Treatment with 32 mg/kg (2R,6R)-HNK significantly reduced withdrawal latency at the 24 hour time point (t(13) = 4.52, P < 0.001; Fig 2).

## Effects of acute dosing on acid-stimulated writhing and acid-depressed rearing

There was a significant main effect of acid for writhing (Fig 3A; F(3, 39) = 32.13, P < 0.001); however, there was no main effect for sex (F(1, 13) = 0.008, NS) or significant interaction (F(3, 39) = 0.07, NS). The Tukey post hoc test revealed that 0.56% acetic acid significantly increased writhing in both sexes as compared to 0%, 0.18%, and 0.32% acetic acid concentrations (P < 0.001). There was an acid concentration-dependent decrease in rearing (Fig 3B; F(3, 39) = 18.56, P < 0.001); however there was no main effect for sex (F(1, 13) = 1.87, NS) or significant interaction (F(3, 39) = 0.70, NS). The Tukey post hoc test revealed that 0.56% acetic acid significantly decreased rearing as compared to 0%, 0.18%, and 0.32% acetic acid concentrations (P < 0.001).

For writhing, administration of 0.56% acetic acid produced a significant increase in writhing that was completely reversed by 3.2 mg/kg ketoprofen (t(14) = 5.16, P < 0.001; Fig 3C). For rearing, 3.2 mg/kg ketoprofen completely reversed the acid-induced decrease of rearing in both male and female mice (t(14) = 15.69, P < 0.001; Fig 3D).

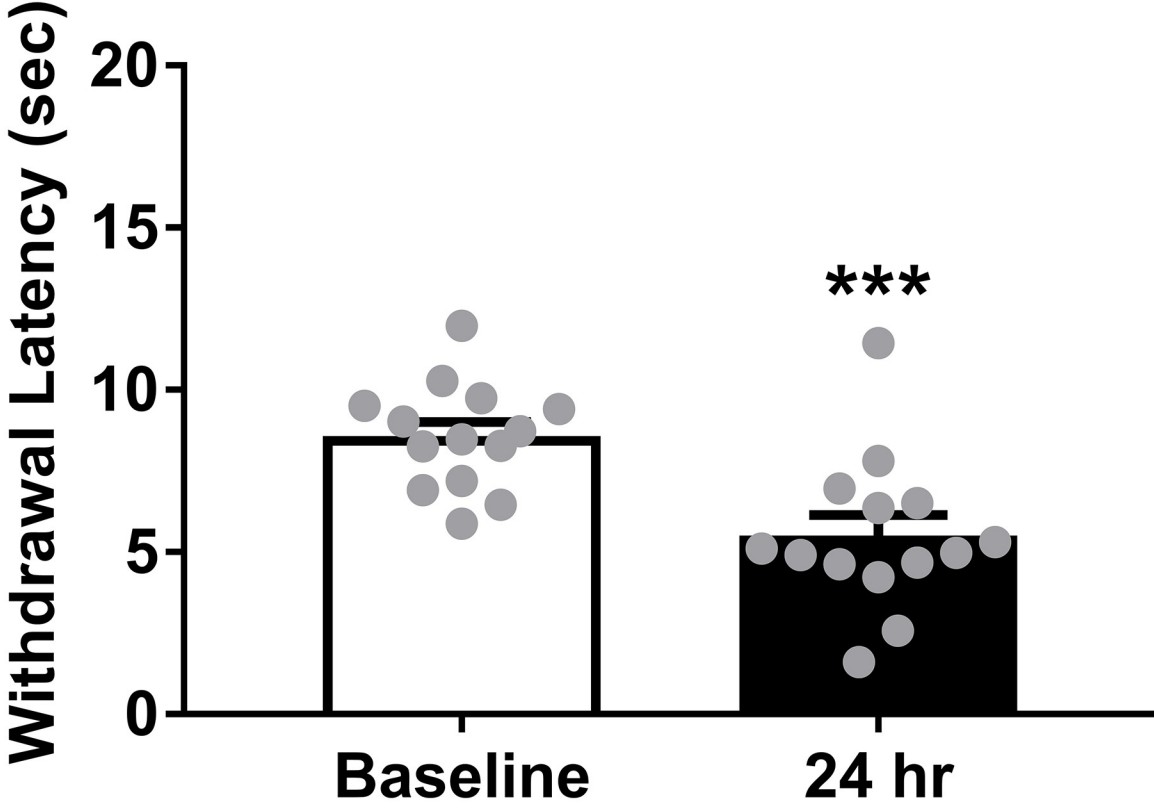

**Fig 2. Effect of 24 hours pretreatment of (2R,6R)-HNK on withdrawal latency in the 56ºC hot plate assay.** (2R,6R)-HNK significantly decreased withdrawal latencies in the 56ºC hot plate. These were the same mice from the Fig 1C tested 24 hours after the final cumulative dose. The Y-axis highest value represents the cutoff time. ***P < 0.001 represents the significant difference from baseline withdrawal latency. All data show mean ± SEM. (Male N = 7, Female N = 7).

Although there was a downward trend, treatment with (2R,6R)-HNK failed to reverse acid-stimulated writhing behavior (F(3, 27) = 0.81, NS; Fig 3E); however, 10 mg/kg (2R,6R)-HNK was found to significantly increase rearing as compared to saline (F(3, 27) = 3.12, P = 0.04; Fig 3F).

### Effects of acute dosing on acid-depressed locomotor activity and rearing

There was a significant main effect of acid for distance traveled (Fig 4A; F(3, 45) = 22.32, P < 0.001); however, there was no main effect for sex (F(1, 15) = 2.99, NS) or significant interaction (F(3, 45) = 0.83, NS). The Tukey post hoc test revealed an acid concentration-dependent effect in which 0.32% and 0.56% acetic acid significantly decreased distance traveled in both sexes as compared to all other acetic acid concentrations (P < 0.001). There was a significant main effect of acid for rearing (Fig 4B; F(3, 45) = 12.64, P < 0.001); however, there was no main effect for sex (F(1, 15) = 0.70, NS) or significant interaction (F(3, 45) = 0.27, NS). The Tukey post hoc test revealed that 0.32% and 0.56% acetic acid significantly decreased rearing as compared to 0% acetic acid concentrations (P < 0.01).

For distance traveled (Fig 4C), 0.56% acetic acid produced a significant decrease in locomotor activity (F(2,32) = 27.10, P < 0.001). A Tukey post hoc test revealed that treatment with 3.2 mg/kg ketoprofen completely reversed acid-depressed locomotor activity in mice. Ketoprofen (3.2 mg/kg) completely reversed acid-depressed rearing (F(2,32) = 21.28, P < 0.001; Fig 4D).

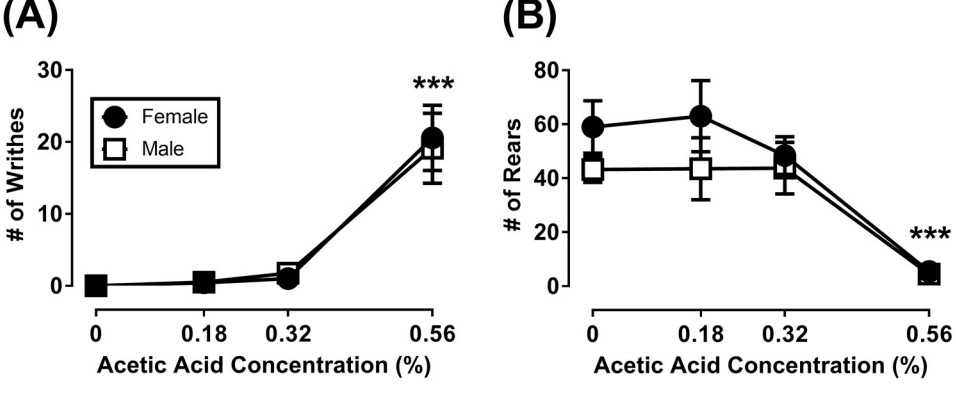

## Ketoprofen

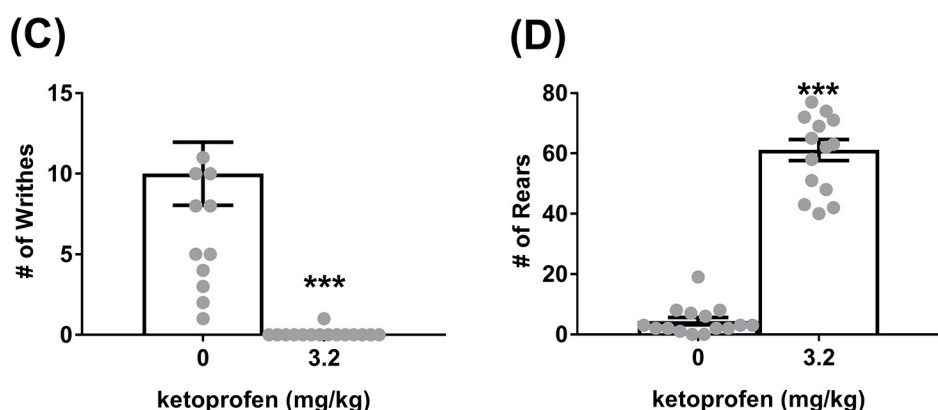

## (2*R*,6*R*)-HNK

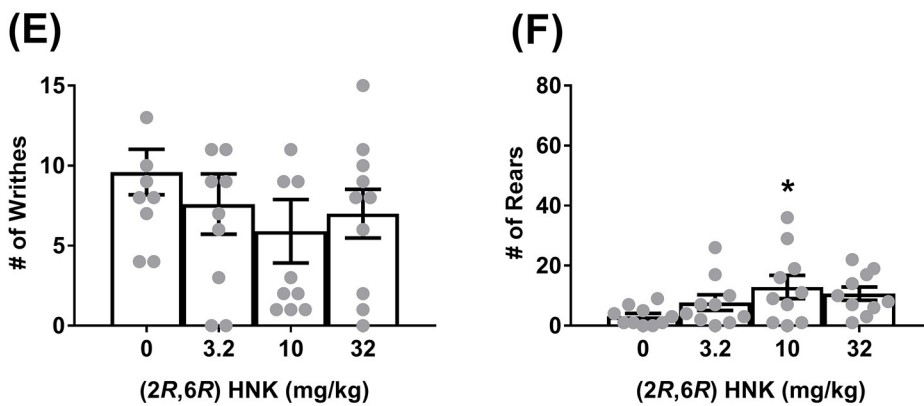

**Fig 3. Effects of sex and acute dosing of (2*R*,6*R*)-HNK on acid-stimulated writhing and acid-depressed rearing.**
(A, B) Shows the effects of acetic acid concentration to induce acid-stimulated writhing and acid-depressed rearing in male and female mice. (C, D) Treatment with ketoprofen completely reversed the effects of acid on writhing and rearing. (E, F) Shows the effects of (2*R*,6*R*)-HNK on writhing and rearing. All drugs were administered 30 mins prior to the start of the experiment. All significant ANOVAs were followed by a Dunnett's post hoc test. *$P < 0.05$, ***$P < 0.001$ represents a significant reversal versus control groups (0mg/kg). +++$P < 0.001$, represent significant

effect versus vehicle + 0.56% acetic acid. All data show mean ± SEM. Grey circles are individual data points. Panel A, B (Male N = 8, Female N = 7); Panel C, D (Male N = 8, Female N = 7); Panel E, F (Male N = 4, Female N = 6).

For the (2*R*,6*R*)-HNK experiments, there was a significant main effect of acid condition (Fig 4E; $F(1, 11) = 46.47$, $P < 0.001$), but no significant interaction ($F(3, 33) = 0.36$, NS), main effect of dose ($F(3, 33) = 1.01$, NS). Treatment with 0.56% acetic acid significantly decreased distance traveled and rearing behavior ($P < 0.001$). Treatment with (2*R*,6*R*)-HNK failed to reverse acid-depressed locomotor activity. For rearing there was a significant main effect of acid condition (Fig 4F; $F(1, 11) = 36.08$, $P < 0.001$), but no significant interaction ($F(3, 33) = 2.05$, N.S), or main effect of dose ($F(3, 33) = 0.96$, NS). (2*R*,6*R*)-HNK failed to reverse acid-depressed changes in rearing behavior.

## Effects of intermittent (2*R*,6*R*)-HNK administration on hot plate latency and acid-depressed LMA

Intermittent administration of 10 mg/kg (2*R*,6*R*)-HNK failed to significantly change withdrawal latency in the 52°C hot plate test as compared to saline (Fig 5A) There was no significant interaction $F(1, 24) = 0.02$, NS), no significant main effect of test condition ($F(1, 24) = 2.16$, NS), and no significant main effect of dose ($F(1, 24) = 0.15$, NS). For the pain-depressed locomotor activity studies (Fig 5B), 0.56% acetic acid significantly decreased distance traveled. There was a significant main effect of acid ($F(2, 32) = 41.48$, $P < 0.001$), no main effect of dose ($F(1, 16) = 2.29$, NS), and no interaction ($F(2, 32) = 0.27$, NS). For rearing (Fig 5C) there was a significant main effect of acid ($F(2, 32) = 65.92$, $P < 0.001$), no main effect of dose ($F(1, 16) = 0.06$, NS), and no interaction ($F(2, 32) = 0.98$, NS)]. Intermittent 10 mg/kg (2*R*,6*R*)-HNK failed to reverse acid-depressed changes in locomotor activity and rearing (Fig 4B and 4C, respectively).

## Discussion

The present study evaluated the antinociceptive effects of (2*R*,6*R*)-HNK in several assays of pain-stimulated and pain-depressed behaviors using both an acute (one injection) and intermittent (4 injections over 6 days) administration. Additionally, we conducted baseline experiments (sex, temperature-curves, and acid-concentrations curves), positive controls to demonstrate that the behavioral assays used in the present study produced reliable results. Consistent with previous studies, the positive controls of morphine (hot plate) and ketoprofen (writhing and pain-depressed locomotor activity) produced maximum antinociceptive effects in their respective assays [14,17,22,23,25,26,30]. These positive controls demonstrated that the pain stimuli used in these assays are reversible and do not require an opioid mechanism (writhing and locomotor activity) as ketoprofen is an NSAID.

In contrast to Yost and colleagues [13], who found that 10 mg/kg (2*R*,6*R*)-HNK increased withdrawal latency in the hot plate assay, the present study found that (2*R*,6*R*)-HNK did not change withdrawal latency at 52°C and decreased withdrawal latency at 56°C (Fig 1C). There are two methodological differences between their study and ours, temperature and pretreatment time, that could contribute to the differences between the two studies. For example, Yost et al. [13] used a 50°C hot plate, which is a lower efficacy requiring assay/temperature as compared to the 52°C and 56°C hot plate tests. Although 10 mg/kg (2*R*,6*R*)-HNK was tested at five different timepoints in the 50°C hot plate assay, (2*R*,6*R*)-HNK only produced an antinociceptive response at one timepoint (24 hours). Interestingly, Goswami and colleagues [9] found

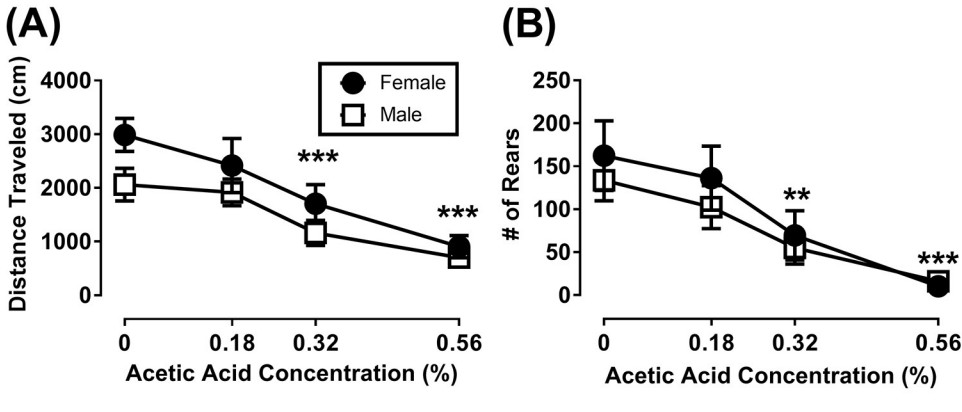

## Ketoprofen

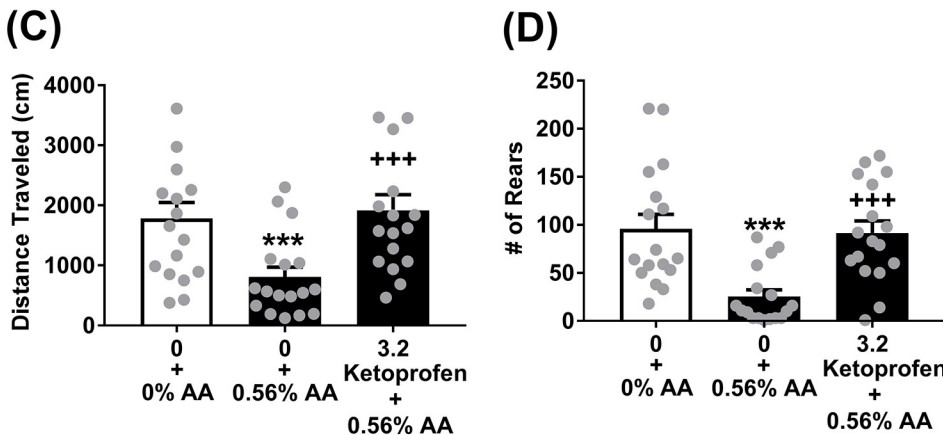

## (2*R*,6*R*)-HNK

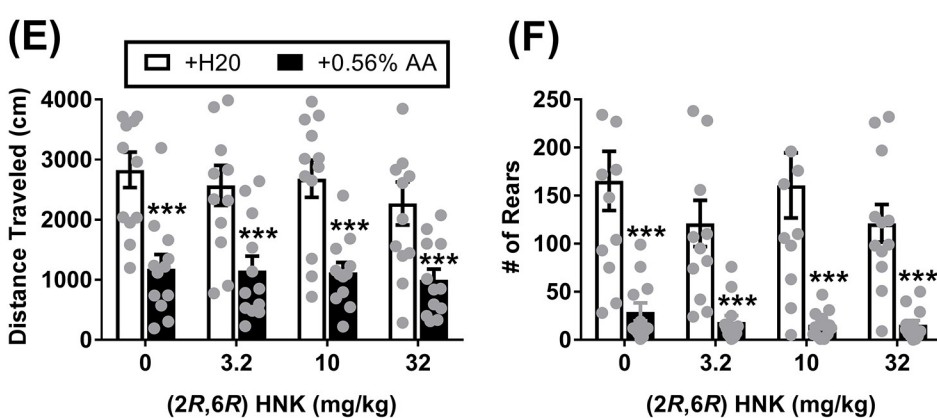

**Fig 4. Effects of sex and acute dosing of (2*R*,6*R*)-HNK on acid-depressed locomotor activity and rearing.** (A, B) Show the effects of acetic acid concentration on distance traveled (locomotor activity) and rearing in male and female mice. (C, D) Treatment with ketoprofen completely reversed acid-depressed locomotor activity and rearing. (E, F) Shows the effects of (2*R*,6*R*)-HNK on pain-depressed locomotor activity and rearing. All drugs were administered 30 mins prior to the start of the experiment. All significant ANOVAs were followed by a Tukey post hoc test. **P < 0.01, ***P < 0.001 represents a significant effect versus control groups. ++P < 0.05, +++P < 0.001, represent significant

reversal versus vehicle + 0.56% acetic acid. All data show mean ± SEM. Grey circles are individual data points. Panel A, B (Male N = 9, Female N = 8); Panel C, D (Male N = 9, Female N = 8); Panel E, F, G, H (Male N = 6, Female N = 6).

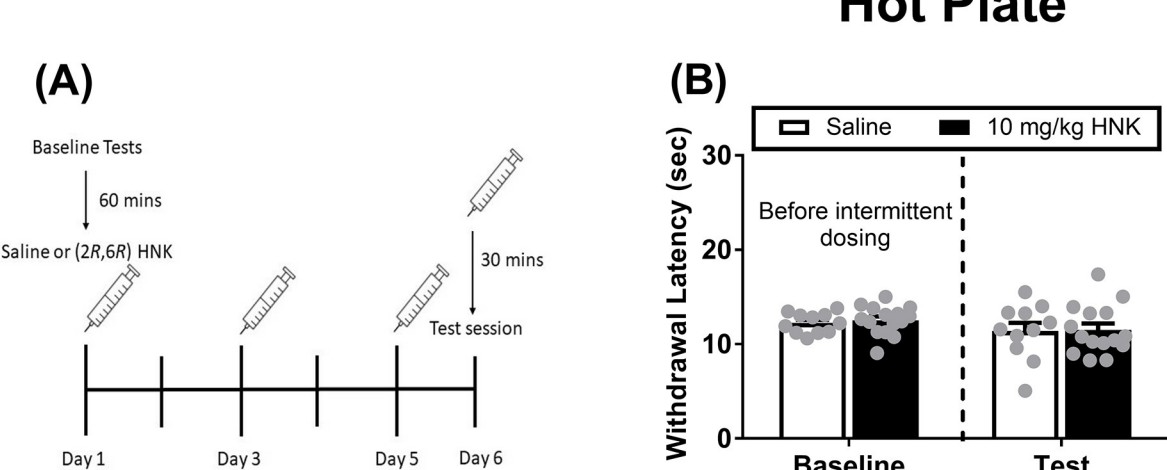

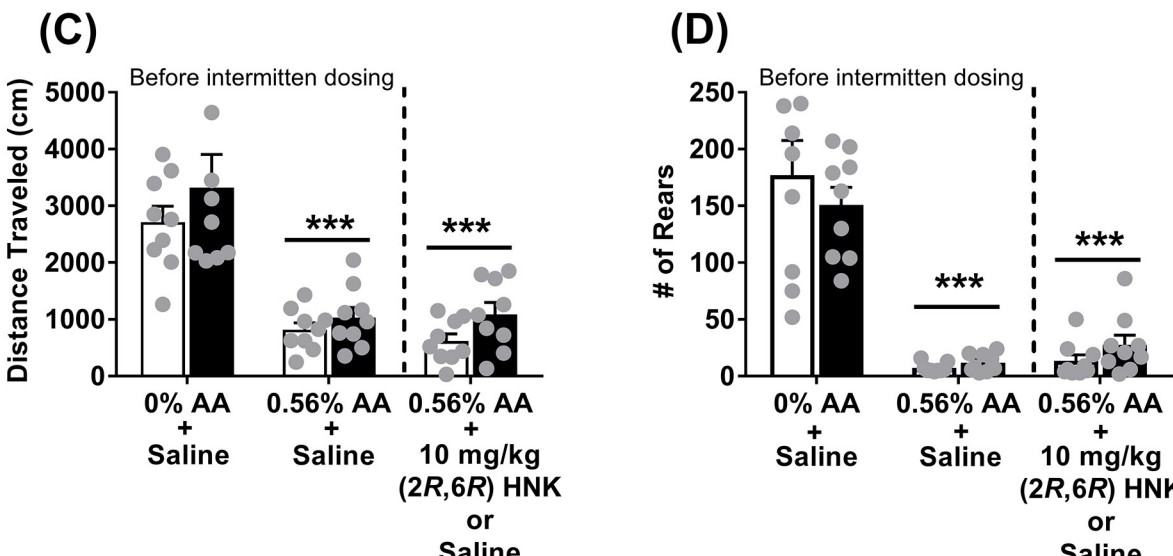

**Fig 5. Effects of intermittent (2R,6R)-HNK administration in 52ºC hot plate and on acid-depressed locomotor activity and rearing.** (A) Provides a visual timeline of injections and behavioral testing. Baseline tests (Saline+0.56% acetic acid and Vehicle+0.56% acetic acid) were conducted prior to the initiation of intermittent dosing. Sixty mins after baseline tests concluded, the first injections were administered. The syringes injection the days in which saline or 10 mg/kg (2R,6R)-HNK were administered. Thirty mins after the final injection on day 6, mice were tested in either the 52ºC hot plate or pain-depressed locomotor activity. (B) Intermittent 10 mg/kg (2R,6R)-HNK did not change withdrawal latencies in the 52ºC hot plate as compared to saline. Baseline withdrawal latency was determined prior to intermittent (2R,6R)-HNK administration. Intermittent (2R,6R)-HNK failed to significantly alter acid-depressed (C) locomotor and (D) rearing. All significant ANOVAs were followed by a Tukey post hoc test. **$P < 0.01$, ***$P < 0.001$ represents a significant effect versus control groups. All data show mean ± SEM. Grey circles are individual data points. Panel A (Saline group: Male N = 7, Female N = 4; HNK group: Male N = 10, Female N = 5); Panel B, C (Saline group: Male N = 6, Female N = 3; HNK group: Male N = 6, Female N = 3). AA = acetic acid.

that intranasal administration of (2R, 6R)-HNK produces a more rapid antinociceptive response in the 52°C hot plate assay. We failed to find antinociceptive effects in the 52°C hot plate with intermittent dosing when (2R,6R)-HNK had been on board for at least 24 hours (4 injections over 6 days). It also is important to note that there is a level of subjectivity in hot plate assays as experimenters are tasked with identifying behaviors, which can be subtle, from a distance to not disrupt the mouse's behavior. The hot plate is not an automated assay, which leaves room for experimenter error. Additionally, it is possible that the size of the hot plate influences behavior as a larger plate allows the mouse more space to move providing competing behavior (thermal response [lick, shake or jump] vs walking/exploring).

(2R,6R)-HNK also failed to produce an antinociceptive effect in pain-depressed behaviors, which is concerning as these assays provide more translational value does than the hot plate assay. Additionally, pain-depressed locomotor activity is a lower efficacy requiring assay as compared to the hot plate assays. For example, 32 mg/kg morphine is required to reach maximum withdrawal latency in the 52°C and 56°C hot plate assays; whereas, 5.6 mg/kg morphine is required to reverse acetic acid-induced decreases in locomotor activity [17]. The pain-depressed locomotor activity assay provides data on two behavioral outcomes (distance traveled and rearing) in the presence and absence of acid. Here, we reported that (2R,6R)-HNK did not alter locomotor activity or rearing in the absence of acid, which is consistent with previous reports [7,8,12,13]. Acute and intermittent administration of (2R,6R)-HNK was unable to reverse the acid-induced depression of locomotor activity and rearing at doses 3 times higher (32 mg/kg) than doses previously found to produce an antinociceptive effect in hot plate and chronic pain state models (10 mg/kg) [10,12,13]. The inability of (2R,6R)-HNK to reverse acid-induced depression of rearing was also found in the writhing assay. The present study analyzed all data with sex as a variable to ensure that a false negative was not the result of sex-related drug effect (e.g. a positive effect in male mice, but a negative or no effect in female mice or vice versa). We did not find a sex effect in any of the temperature or acid-concentration baseline/control experiments. Moreover, there was no indication of sex-related drug effects following (2R,6R)-HNK administration (see supplementary materials).

(2R,6R)-HNK has shown sustained antinociceptive effects in models of chronic inflammatory pain states including complex regional pain syndrome type-1, plantar incision postoperative pain, spared nerve injury, and mechanical hypersensitivity produced by carrageenan in the hind paw [10,12,13]. Interestingly, the half-life of (2R,6R)-HNK in brain and plasma is less than one hour, and completely eliminated by the onset or during the sustained antinociceptive effects [5,7]. The sustained antinociceptive effects in these chronic inflammatory pain models, but not acute pain models using thermal or acid pain states, indicates that (2R,6R)-HNK is producing long-term changes in the inflammatory response and/or neuroplasticity. Although we do not know the mechanism by which (2R,6R)-HNK produces these long-lasting changes, Yost and colleagues [12,13] have shown that the α-amino-3-hydroxy-5-methyl-4-isoxazole-propionic acid (AMPA) receptors, but not opioid receptors, are required for the antinociceptive effects. If the opioid receptors do not play any role in the antinociceptive effects of (2R,6R)-HNK, this would be an important finding, as (2R,6R)-HNK would provide a safer option than current opioids. In addition, (2R,6R)-HNK is devoid of abuse-related and psychomimetic effects that have been found with ketamine and opioid-based drugs [7,8].

## Other considerations

The present study was unable to replicate the positive effects of (2R,6R)-HNK in the hot plate assays reported by Yost and colleagues [13]. We have focused on this report because it used the same acute pain assay, while the other papers all used chronic/inflammatory pain states, which

are very different from a mechanistic standpoint. There were several similarities between the two studies including the supplier of (2*R*,6*R*)-HNK (NCATS), strain of mice (C57BL/6), and behavioral outcomes (latency to jump or lick). There were also several differences. For example, mouse strains were secured from different vendors (Envigo vs Jackson Laboratories) and hot plate temperature (52ºC and 56ºC vs 50ºC), which can influence behaviors (see discussion above on hot plate assay). We use best practices in terms of establishing behavioral baselines for each assay, using appropriate, positive controls and removing odors with ethanol that can influence behavior. Unfortunately, there are many other variables that can influence behaviors, and it can be very difficult to control all of these variables during the data collection process. The present study did not control for the estrous stage, housing male and female mice in different locations, or sex of the experimenter. We are a small undergraduate university with limited research space, thus making it difficult to control these variables. However, if these variables are responsible for the lack of antinociceptive effects of (2*R*,6*R*)-HNK in thr present, it is unlikely that (2*R*,6*R*)-HNK will be successful in the clinical population as an acute pain medication. Lastly, and most importantly, there is a documented history of (2*R*,6*R*)-HNK producing null effects (i.e. failing to produce significant effects) in assays used to measure antidepressant effects (See our commentary for more details [31–35].

## Conclusion

In conclusion, we found that (2*R*,6*R*)-HNK failed to produced acute antinociceptive effects in the assays of pain-stimulated and pain-depressed behaviors tested in the present study. These results caution the development of (2*R*,6*R*)-HNK as short-term, quick acting analgesic due to what appears to be a limited application; that is, only one reported antinociceptive effect in one assay (50ºC hot plate test) with a 24-hour pretreatment time. However, there is promise that (2*R*,6*R*)-HNK can be developed for chronic inflammatory pain states. Currently, all of the successful preclinical chronic pain studies with (2*R*,6*R*)-HNK have used pain-stimulated behaviors. Future studies need to evaluate the effects of (2*R*,6*R*)-HNK on pain-depressed behaviors using a chronic pain state like formalin or repeated acid to depress intracranial self-stimulation (ICSS), locomotor activity, or wheel running [27,28,36–38]. However, it should be noted that (2*R*,6*R*)-HNK does not appear to share any pharmacological profile similarities with ketamine and it appears to lack affinity for the receptors that ketamine binds to in the brain [39]. Based on this, Bonaventura et al. [39] postulate that if (2*R*,6*R*)-HNK is found to be useful for the treatment of chronic pain, it should "lack the abuse liability and other negative side-effects associated with ketamine and (*S*)-ketamine."

## Supporting information

**S1 File.**
(PDF)

**S2 File.**
(DOCX)

**S3 File.**
(DOCX)

## Author Contributions

**Conceptualization:** Todd M. Hillhouse, Joseph H. Porter.

**Data curation:** Todd M. Hillhouse, Kaitlyn J. Partridge, Patrick I. Garrett, Sarah C. Honeycutt.

**Formal analysis:** Todd M. Hillhouse.

**Investigation:** Todd M. Hillhouse, Kaitlyn J. Partridge, Patrick I. Garrett, Sarah C. Honeycutt.

**Methodology:** Todd M. Hillhouse, Kaitlyn J. Partridge, Patrick I. Garrett, Joseph H. Porter.

**Resources:** Todd M. Hillhouse.

**Validation:** Todd M. Hillhouse.

**Writing – original draft:** Todd M. Hillhouse, Kaitlyn J. Partridge, Patrick I. Garrett, Sarah C. Honeycutt.

**Writing – review & editing:** Todd M. Hillhouse, Kaitlyn J. Partridge, Joseph H. Porter.

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
