## [Decision Letter · Decision Letter 0]

16 Jan 2024

PONE-D-23-36070(2R,6R)-hydroxynorketamine fails to produce antinociception in assays of acute pain-stimulated and pain-depressed behaviors in micePLOS ONE

Dear Dr. Porter,

Thank you for submitting your manuscript to PLOS ONE. After careful consideration, we feel that it has merit but does not fully meet PLOS ONE’s publication criteria as it currently stands. Therefore, we invite you to submit a revised version of the manuscript that addresses the points raised during the review process.

We look forward to receiving your revised manuscript.

Kind regards,

Onder Tutsoy

Academic Editor

PLOS ONE

3. As part of your revision, please complete and submit a copy of the Full ARRIVE 2.0 Guidelines checklist, a document that aims to improve experimental reporting and reproducibility of animal studies for purposes of post-publication data analysis and reproducibility: https://arriveguidelines.org/sites/arrive/files/documents/Author%20Checklist%20-%20Full.pdf Please include your completed checklist as a Supporting Information file. Note that if your paper is accepted for publication, this checklist will be published as part of your article.

4. In the online submission form you indicate that your data is not available for proprietary reasons and have provided a contact point for accessing this data. Please note that your current contact point is a co-author on this manuscript. According to our Data Policy, the contact point must not be an author on the manuscript and must be an institutional contact, ideally not an individual. Please revise your data statement to a non-author institutional point of contact, such as a data access or ethics committee, and send this to us via return email. Please also include contact information for the third party organization, and please include the full citation of where the data can be found.

Reviewers' comments:

Reviewer's Responses to Questions

**Comments to the Author**

1. Is the manuscript technically sound, and do the data support the conclusions?

Reviewer #1: Yes

Reviewer #2: No

2. Has the statistical analysis been performed appropriately and rigorously? 

Reviewer #1: Yes

Reviewer #2: No

3. Have the authors made all data underlying the findings in their manuscript fully available?

Reviewer #1: Yes

Reviewer #2: Yes

4. Is the manuscript presented in an intelligible fashion and written in standard English?

Reviewer #1: Yes

Reviewer #2: Yes

5. Review Comments to the Author

Reviewer #1: The study demonstrated that (2R,6R)-hydroxynorketamine fails to produce antinociception even when it was applied as three intermittent doses. Since study is contradictory to several earlier findings, it is suggested to check the quality and reliability of the HNK molecules used for this study. Authors to provide the detail of the (2R,6R)-hydroxynorketamine like catalogue number, purity and grade.

It is suggested that at least hot plate test should be carried out using HNK molecule from three different source to rule out the quality issue.

Reviewer #2: Hillhouse et al report a series of studies evaluating the ketamine metabolite (2R,6R)-hydroxynorketamine, for potential antinociceptive properties in response to thermal stimuli (hot plate test) and attenuation of acetic acid withing behavior. They also screen the compound in restore rearing behavior, an operationally defined pain-depressed behavior evoked by the acetic acid administration. The rationale behind the studies is primarily to extend the current literature to include acute pain studies, as the literature predominately reports the efficacy of this compound in rodent models of chronic pain. This rationale is weak given that at least two other published articles have reported antinociception using the hot plate Yost et al 2022, JPET (intraperitoneal administration) and Goswami et al., 2023, CEPP, (intranasal administration) and acute inflammatory pain models, formalin (Goswami 2023) and carrageenan (Yost 2022, Neuropharmacology). However, given the relatively few reports in the literature, the content of the manuscript has merit and should be of interest to the field. Below are comments that the authors should consider addressing.

Major concerns

Study Design: While endeavoring to include both sexes, the variability of the dataset has weakened the ability of the authors to detect meaningful changes with such a low n per sex per group. Please provide the power analysis for each the reported studies. The variance is high given the sex differences between the two studies, with more consistent responses between sexes in the ketoprofen study and indeed greater variation between the sexes in the HNK study relative potentially due to the Latin Square Design. Moreover, based on the literature, females exhibit behavioral response in response to higher doses that those required by male mice. Given the variance in the model at baseline in terms of sexes it seems like it is not sufficiently powered here for the lower doses given the variance in responses between the sexes. This would lead to the conclusion that greater numbers are required to equivocally assess the outcomes.

Acetic Acid Writhing: Were those animals used or the ketoprofen study only exposed to acetic acid once?

The use of the Latin square design for (2R,6R)-hydroxynorketamine dose response curve with the acetic acid writhing test. While I appreciated the use of the design to limit the number of animals required, it is questionable for this assay. It is unclear whether the same animals were tested weekly for the acetic acid writhing test or whether the (2R,6R)-HNK doses were administered within a specific timeframe following the first acetic acid writhing induction paradigm. If acetic acid was administered repeatedly, this is no longer an acute inflammatory insult with repeated inductions. Did you observe sensitization to the acetic acid in these animals and hence induce a more chronic inflammatory condition, or greater pain related responses in either sex?

In addition, the logistics of the “reversal test” do not allow for post writhing induction treatment. All of these evaluations are prophylactic studies.

Hot plate Assay: Please confirm whether “first paw lick” refers to the hindpaws, forepaws or any paw? Did you have a different behavioral profile of thermal responses in male and female mice? Did the raters always agree on the first response?

Time of testing post treatment: In the reported literature, testing time points post treatment for intraperitoneal dosing indicate the onset of drug effects analgesia and antinociception as occurring 2-4 hours post treatment, with the optimal effects observed at 24 h post treatment. A similar profile for intrathecal administration has been reported, with intranasal (2R,6R)-HNK resulting in more rapid activity, with significant antinociception on the hot plate at 30- and 60-minutes following treatment. Similarly, for the intermittent dosing study, the authors reference Wulf et al., which again examined the efficacy of (2R<6R)-HNK at 24 h following the final treatment. Thus, the time points of testing used in this manuscript (30 minutes post treatment) are potentially to early to yield significant effects.

Statistical Analysis: Please do not include Tukey multiple comparison test when there were no significant interactions. If you do wish to look for simple main effects, then you need to modify your post hoc test and do not use a Tukey comparison.

Why not just run a three-way ANOVA to evaluate Sex X Experimental condition X Dose? Or indeed a repeated two way ANOVA for the hot plate studies.

Studies not referenced: The authors should look at the following articles for completeness

Goswami N, Aleem M, Manda K. Intranasal (2R, 6R)-hydroxynorketamine for acute pain: Behavioural and neurophysiological safety analysis in mice. Clin Exp Pharmacol Physiol. 2023; 50(2): 169-177. doi:10.1111/1440-1681.13737

Liu, AR., Lin, ZJ., Wei, M. et al. The potent analgesia of intrathecal 2R, 6R-HNK via TRPA1 inhibition in LF-PENS-induced chronic primary pain model. J Headache Pain 24, 141 (2023). https://doi.org/10.1186/s10194-023-01667-1

Graphical representation of data:

For all of the study designs and the figures, please indicate the time post treatment at which the drugs were evaluated.

Figures are quite pixelated graphs.

Figure 2 - the word (2R,6R)-HNK in the middle of the graph is not type set correct.

The data representation does not always match the data analysis performed. The overall presentation is quite confusing, particularly for the acetic acid writhing and rearing tests.

Minor comments:

The use of the term “pain-stimulated” is perhaps not the quite right. For the hot plate test this is really a noxious thermal stimulus, or simply a test of nociception rather than pain per se, which is a very human based concept. Acetic acid writhing is again a behavioral model that relies on an inflammatory response and the activation of nociceptors. It is always difficult to determine the most meaningful phrasing without over reaching and using terms that are not quite accurate for the conditions under evaluation.

The validity of rearing in the open field as a pain depressed behavior is somewhat controversial. Is this truly a pain depressed behavior as we know it in the context of human pain. It would be more useful to have some active engagement rather than passive exploration in the open field. This is perhaps more relevant in those chronic rodent models, where the use of limb is downgrading to prevent pain, or indeed that motor activity is reduced to limit the associated “pain”. There are other models that do require functional activities, nesting, food gathering, or step-down tests/incline test etc that also better explore functionality. Moreover, I would find it difficult to support a medication that induces hyperactivity rather than a restoration of normal function.

Is it more beneficial to have a compound that produces overall anti-nociception rather than just analgesia alone. I would not be happy with a compound that modulated this behavior.

In the discussion when referring to the differences between the current and previously published methods, please also discuss the differences in apparatus for the hot plates used and the level of subjectivity to these assays. Now as more reports are emerging that use more automated recording of responses this should improve with reports. In addition, the intranasal study dose shows significant antinociception on the hot plate within the hour of administration - again with a much higher n that included here for one sex only.

6. PLOS authors have the option to publish the peer review hi

---

## [Author Response · Author response to Decision Letter 0]

28 Feb 2024

Response to Reviewers

Reviewer #1: 

The study demonstrated that (2R,6R)-hydroxynorketamine fails to produce antinociception even when it was applied as three intermittent doses. Since study is contradictory to several earlier findings, it is suggested to check the quality and reliability of the HNK molecules used for this study. Authors to provide the detail of the (2R,6R)-hydroxynorketamine like catalogue number, purity and grade.

It is suggested that at least hot plate test should be carried out using HNK molecule from three different source to rule out the quality issue.

RESPONSE: We have contacted the National Center for Advancing Translational Sciences (NCATS) to confirm the purity of the compounds. They have provided us with a certificate of analysis that we have included in the supplemental materials. Additionally, we have added the following information to the drug sections (Page 6, Lines 120-121) “(2R,6R)-HNK was provided by the National Center for Advancing Translational Sciences (NCATS; Bethesda, Maryland, USA; purity >99.5%, certification of analysis is available in the supplementary materials).”

Although we understand why the reviewer suggested that we test 3 different sources of HNK to rule out quality issues, we have confirmed the purity and provided the required documentation. It should be noted that NCATS is the chemist group that provides (2R,6R)-HNK to the Gould research group, which has published the most on the antidepressant effects of (2R,6R)-HNK (Zanos et al., 2019). Additionally, we feel this would not be a great use of animals from an ethical standpoint and does not align with ARRIVE or the 3 R’s. Lastly, and most importantly, there is a history of (2R,6R)-HNK producing negative results (no significant effects) (Shirayama et al., 2018; Xiong et al., 2019; Yang et al., 2017; Zhang et al., 2018). In fact, the British Journal of Pharmacology invited us to write a commentary on the replication issues of HNK (Hillhouse et al., 2019).

Hillhouse, T., Rice, R., & Porter, J. (2019). What role does the ( 2R,6R )‐hydroxynorketamine metabolite play in the antidepressant‐like and abuse‐related effects of ( R )‐ketamine? British Journal of Pharmacology, 176. doi:10.1111/bph.14785

Shirayama, Y., & Hashimoto, K. (2018). Lack of Antidepressant Effects of (2R,6R)-Hydroxynorketamine in a Rat Learned Helplessness Model: Comparison with (R)-Ketamine. Int J Neuropsychopharmacol, 21(1), 84-88. doi:10.1093/ijnp/pyx108

Xiong, Z., Fujita, Y., Zhang, K., Pu, Y., Chang, L., Ma, M., . . . Hashimoto, K. (2019). Beneficial effects of (R)-ketamine, but not its metabolite (2R,6R)-hydroxynorketamine, in the depression-like phenotype, inflammatory bone markers, and bone mineral density in a chronic social defeat stress model. Behavioural Brain Research, 368, 111904. doi:https://doi.org/10.1016/j.bbr.2019.111904

Yang, C., Qu, Y., Abe, M., Nozawa, D., Chaki, S., & Hashimoto, K. (2017). (R)-Ketamine Shows Greater Potency and Longer Lasting Antidepressant Effects Than Its Metabolite (2R,6R)-Hydroxynorketamine. Biol Psychiatry, 82(5), e43-e44. doi:10.1016/j.biopsych.2016.12.020

Zhang, K., Fujita, Y., & Hashimoto, K. (2018). Lack of metabolism in (R)-ketamine’s antidepressant actions in a chronic social defeat stress model. Scientific Reports, 8(1), 4007. doi:10.1038/s41598-018-22449-9

Zanos, P., Highland, J. N., Stewart, B. W., Georgiou, P., Jenne, C. E., Lovett, J., . . . Gould, T. D. (2019). (2R,6R)-hydroxynorketamine exerts mGlu2 receptor-dependent antidepressant actions. Proceedings of the National Academy of Sciences, 116(13), 6441. doi:10.1073/pnas.1819540116

Reviewer #2: 

Hillhouse et al report a series of studies evaluating the ketamine metabolite (2R,6R)-hydroxynorketamine, for potential antinociceptive properties in response to thermal stimuli (hot plate test) and attenuation of acetic acid withing behavior. They also screen the compound in restore rearing behavior, an operationally defined pain-depressed behavior evoked by the acetic acid administration. The rationale behind the studies is primarily to extend the current literature to include acute pain studies, as the literature predominately reports the efficacy of this compound in rodent models of chronic pain. This rationale is weak given that at least two other published articles have reported antinociception using the hot plate Yost et al 2022, JPET (intraperitoneal administration) and Goswami et al., 2023, CEPP, (intranasal administration) and acute inflammatory pain models, formalin (Goswami 2023) and carrageenan (Yost 2022, Neuropharmacology). However, given the relatively few reports in the literature, the content of the manuscript has merit and should be of interest to the field. Below are comments that the authors should consider addressing.

RESPONSE: To be fully transparent, we started collecting the data for this manuscript in fall 2019, but we had to close the lab due to Covid restrictions in early 2020. At the time we started collecting data, there was only one publication that evaluated the effects of HNK on chronic pain (Kroin et al., 2018). Once Covid restrictions were lifted we started collecting data again, then we found the publications by Lucki’s research group (Yost et al., 2022). Based on the findings in the Yost et al., (2022) article, we added the intermittent dosing data to rule out the pretreatment time/dosing (30 min vs 24 hour) and to expand the findings.

We feel we have provided a rationale for our studies in the introduction. We have added text to further support the need to test HNK in assays of pain-depressed behavior as these provide a translational value that is missing from other studies. 

Pages 3-4, Lines 69-95: “To date, (2R,6R)-HNK has only been evaluated in assays of pain-stimulated behaviors, which exclusively rely on behaviors that increase in rate, intensity, and frequency following the presentation or administration of a noxious stimulus (Hillhouse and Negus, 2016; Negus et al., 2010; Negus, 2019). Pain-stimulated behaviors can be attenuated/reversed by drugs that block or reduce the sensitivity to the noxious stimulus OR by drugs that produce motor sedation or impair general locomotor activity (e.g. haloperidol or kappa agonists). Although results from pain-stimulated studies are promising and can provide valuable information about candidate analgesics, complete reliance on assays of pain-stimulated behavior may prove to be problematic leading to false positives (impair motor function as described above). Assays of pain-depressed behaviors can be defined as behaviors that decrease in rate, intensity, or frequency following to administration/presentation of a noxious stimulus and used alone or to complement assays of pain-stimulated behavior (Negus et al., 2010; Negus et al., 2019; Stevenson et al., 2009). These pain-depressed behavioral assays provide a translation aspect that is missing with pain-stimulated behavior, as people dealing with pain typically decrease their activity and work productivity (Kawai et al., 2017). Additionally, pain-depressed assays are able to identify compounds that produce false positives of pain-stimulated due to motor impairment as these compounds would not be able increase behaviors back to baseline levels. In the present study, we used an assays of pain-depressed locomotor activity as both acetic and lactic acid produce a robust depression of locomotor activity that can be reversed with nonsteroidal anti-inflammatory drug (NSAID) as well as standard opioid drugs (Negus et al., 2023; Stevenson et al., 2009). In addition to determining if a drug can reverse the acid-induced decrease in locomotor activity and rearing, this assay allows us to evaluate the effects of each drug on locomotor activity in the absence of acid. Specifically, the combination of drug + DH2O (drug alone) is used to rule out false positives as an increase in locomotor activity in the absence and presence of the acetic acid indicated a nonselective increase in locomotor activity not an antinociceptive effect, Additionally, a decrease in locomotor activity in the absence and presence of the acetic acid is a nonselective suppression of behavior resulting in a false negative.”

Major concerns

Study Design: While endeavoring to include both sexes, the variability of the dataset has weakened the ability of the authors to detect meaningful changes with such a low n per sex per group. Please provide the power analysis for each the reported studies. The variance is high given the sex differences between the two studies, with more consistent responses between sexes in the ketoprofen study and indeed greater variation between the sexes in the HNK study relative potentially due to the Latin Square Design. Moreover, based on the literature, females exhibit behavioral response in response to higher doses that those required by male mice. Given the variance in the model at baseline in terms of sexes it seems like it is not sufficiently powered here for the lower doses given the variance in responses between the sexes. This would lead to the conclusion that greater numbers are required to equivocally assess the outcomes.

RESPONSE: We completed a sex and temperature/acid concentration curve for all experiments (Hot plate, fig 1a; writhing, fig3a; locomotor activity/rearing, fig 4a,b) and did not find any significant sex differences under baseline conditions. Additionally, the sex data for the HNK studies all trend in the same direction. Based on these results and the comment above, we have decided to move all of the sex effect graphs (from original submission) to the supplementary materials. We provided the results (statistical values) in the supplementary materials to allow researchers to examine that data as needed. 

The data have been reanalyzed using a one-way repeated measures ANOVA with the male and female data combined together. Significant ANOVAs were followed by Tukey multiple comparison tests. It is important to note that this change in analysis did not change the overall outcome of the studies or manuscript.

Acetic Acid Writhing: Were those animals used or the ketoprofen study only exposed to acetic acid once?

The use of the Latin square design for (2R,6R)-hydroxynorketamine dose response curve with the acetic acid writhing test. While I appreciated the use of the design to limit the number of animals required, it is questionable for this assay. It is unclear whether the same animals were tested weekly for the acetic acid writhing test or whether the (2R,6R)-HNK doses were administered within a specific timeframe following the first acetic acid writhing induction paradigm. If acetic acid was administered repeatedly, this is no longer an acute inflammatory insult with repeated inductions. Did you observe sensitization to the acetic acid in these animals and hence induce a more chronic inflammatory condition, or greater pain related responses in either sex?

RESPONSE: We apologized for any confusion in the design. We used a within-subject design for all of the studies (expect the intermittent dosing studies). We separated acid administration by one week to ensure tolerance or sensitization did not develop. This is a common method used by the Negus research group (Hillhouse and Negus, 2016; Kwilasz et al., 2012; Kwilasz et al., 2014; Pereira Do Carmo et al., 2009; Leitl et al., 2014; Negus et al., 2010). Additionally, the Negus research group has shown that daily acid administration does not produce tolerance or sensitization (Legakis et al., Miller et al., 2015). We monitor mice daily for any pain and distress for mice receiving once a week acid administration, and did not see any noticeable change in behavior. Additionally, we monitor mice for any exaggerated or lack of response to acid on test days and did not see a change in acid effects.

We have edited the text in the writhing and pain-depressed locomotor activity section to clarify when and how drugs and acid was administered. (Page 8, Lines 163-170) “For the drug experiments, (2R,6R)-HNK (0-32 mg/kg) and the positive control ketoprofen (0 or 3.2 mg/kg) were administered 30 mins prior to 0.56% acetic acid administration (which was selected based on the results from the acetic acid concentration curve experiment). Drug doses were randomized using a Latin-square design. There was a minimum of 48 hours between test sessions, and acetic acid administration was separated by a week. For example, on Monday mice were injected with a drug + 0.56% acetic acid, and on Thursday mice were injected with a drug + DH2O. Separate groups of mice were used for (2R,6R)-HNK (Male N = 4; Female N = 6) and ketoprofen (Male N = 8; Female N = 7).

(Page 9, Line 185-193)” For the drug experiments, (2R,6R)-HNK (0-32 mg/kg) and ketoprofen (0 or 3.2 mg/kg) were administered 30 mins prior to 0.56% acetic acid or vehicle (deionized water) injections. Acid vehicle tests were used to determine if (2R,6R)-HNK or ketoprofen could reverse acetic acid depression of LMA without stimulating locomotor activity. Drug doses were randomized using a Latin-square design. To be consistent with the writhing experiments, there was a minimum of 48 hours between test sessions, and acetic acid administration was separated by a week. For example, on Monday mice were injected with a drug + 0.56% acetic acid, and on Thursday mice were injected with a drug + DH2O. Separate groups of mice were used for (2R,6R)-HNK (Male N = 9; Female N = 8) and ketoprofen (Male N = 6; Female N = 6).

In addition, the logistics of the “reversal test” do not allow for post writhing induction treatment. All of these evaluations are prophylactic studies.

RESPONSE: We have removed the phrase “For the drug reversal experiments” (line 163 and 185) and replaced with “For the drug experiments”. The quotes in the previous response show that “reversal” was removed.

Hot plate Assay: Please confirm whether “first paw lick” refers to the hindpaws, forepaws or any paw? Did you have a different behavioral profile of thermal responses in male and female mice? Did the raters always agree on the first response?

RESPONSE: We have edited the text to state “first forepaw or hindpaw lick”. Figure 1a shows the effects differences for the hot plate temperature-curve. We found no sex differences in the temperature-curve OR on baseline values for the animals treated with drug (Figure 1b and 1c). Our raters were very reliable. In the rare event that both raters did not stop their stopwatch at the same time, the experiment continued until the second rater saw a behavioral response (paw lick, paw shake, or jump). Using the average time of the two raters for all mice help account for any differences in rater’s times. We found this to be a better option than relying on a single rater.

Time of testing post treatment: In the reported literature, testing time points post treatment for intraperitoneal dosing indicate the onset of drug effects analgesia and antinociception as occurring 2-4 hours post treatment, with the optimal effects observed at 24 h post treatment. A similar profile for intrathecal administration has been reported, with intranasal (2R,6R)-HNK resulting in more rapid activity, with significant antinociception on the hot plate at 30- and 60-minutes following treatment. Similarly, for the intermittent dosing study, the authors reference Wulf et al., which again examined the efficacy of (2R<6R)-HNK at 24 h following the final treatment. Thus, the time points of testing used in this manuscript (30 minutes post treatment) are potentially to early to yield significant effects.

RESPONSE: As mentioned above, a significant portion of the data reported in our manuscript were collected prior to the publications that provide the 24hr timepoint. Intranasal HNK administration produce antinociceptive effects at all time points test (15, 30, and 60 mins). I would like to point out that these effects were very small (increase of 5 sec at the max effect) with an N = 10. 

We agree that the pretreatment time could be playing a role which is why it is discussed in the discussion (Lines 320-330). Additionally, we added the intermittent dosing studies to address the issue. The mice received 4 injections over 6 days (injections on days 1, 3, 5, and the last injection was 30 mins before the test on d

---

## [Decision Letter · Decision Letter 1]

5 Mar 2024

PONE-D-23-36070R1(2R,6R)-hydroxynorketamine fails to produce antinociception in assays of acute pain-stimulated and pain-depressed behaviors in micePLOS ONE

Dear Dr. Porter,

Thank you for submitting your manuscript to PLOS ONE. After careful consideration, we feel that it has merit but does not fully meet PLOS ONE’s publication criteria as it currently stands. Therefore, we invite you to submit a revised version of the manuscript that addresses the points raised during the review process.

We look forward to receiving your revised manuscript.

Kind regards,

Onder Tutsoy

Academic Editor

PLOS ONE

Reviewers' comments:

Reviewer's Responses to Questions

**Comments to the Author**

1. If the authors have adequately addressed your comments raised in a previous round of review and you feel that this manuscript is now acceptable for publication, you may indicate that here to bypass the “Comments to the Author” section, enter your conflict of interest statement in the “Confidential to Editor” section, and submit your "Accept" recommendation.

Reviewer #1: (No Response)

Reviewer #2: All comments have been addressed

2. Is the manuscript technically sound, and do the data support the conclusions?

Reviewer #1: Partly

Reviewer #2: Yes

3. Has the statistical analysis been performed appropriately and rigorously? 

Reviewer #1: I Don't Know

Reviewer #2: No

4. Have the authors made all data underlying the findings in their manuscript fully available?

Reviewer #1: Yes

Reviewer #2: Yes

5. Is the manuscript presented in an intelligible fashion and written in standard English?

Reviewer #1: Yes

Reviewer #2: Yes

6. Review Comments to the Author

Reviewer #1: Several studies by Jaclyn N. Highland and his team, including those with a comprehensive pharmacokinetics and pharmacodynamics of the HNKs are available suggesting the beneficial effect of HNK. Therefore, current contradictory finding is justifiable only if sufficient data has to be presented to rebut it mechanistically. If not, at least thermal nociception has to be assessed for HNK from three independent sources.

In the current study, authors have to revisit their assays of behavioral phenotyping and the quality of HNK. The quality certificate provided by authors is indicative of “quality at source” but not the “quality at bench”.

Please also check a technically robust paper on even the oral bioavailability of HNK in Mouse, rat, and dog .

Highland, Jaclyn N., et al. "Mouse, rat, and dog bioavailability and mouse oral antidepressant efficacy of (2R, 6R)-hydroxynorketamine." Journal of Psychopharmacology 33.1 (2019): 12-24.

Another study published in the FASEB journal suggest the Analgesic and Antinociceptive Effects of (2R,6R)-hydroxynorketamine (HNK) in Mice even after 24 hrs of drug administration(https://doi.org/10.1096/fasebj.2022.36.S1.R2181 )

Behavioral phenotypes of rodents may also be affected due to several technical, state and trait factors.

In the current study, both the male and female mice were used for assays .The presence of the opposite gender mice in the same housing, test arena and the odor/scent marks of opposite gender is sufficient to perturb the expected behavioral response. Therefore, it is suggested to describe the procedure employed to rule out such confounding factors. Such perturbations are more likely if neighboring mice are in estrous stage.

Following relevant paper of Nature Neuroscience is an example to explain that how important are the state and trait factors in pre-clinical behavioral pharmacology.

Georgiou, Polymnia, et al. "Experimenters’ sex modulates mouse behaviors and neural responses to ketamine via corticotropin releasing factor." Nature neuroscience 25.9 (2022): 1191-1200.

Reviewer #2: Thank you for addressing the comments and including the additional data. Overall, the manuscript is a very interesting report and important for the field. There remain some edits that should be considered for clarity.

With regards to the statistical analysis, following a One -way ANOVA for this data set it is more appropriate to utilize a Dunnett multiple comparison test, with comparison made only to the control group 0 mg/kg rather than exploring differences between all of the doses used. Please make this change.

Figure 1, 3c - the legend indicating male has an error that needs to be corrected.

It remains difficult to compare the ketoprofen data with that of HNK as figure 4e and 4f are presented differently to 4 c and 4d, I realize this is due to the dose response curve but it would make the point more salient if you could graph this as histograms as per 4c and 4 d

In addition, I'm not sure whether it is appropriate to separate the ketoprofen data by sex when you have decided to collapse the data across the sexes for the HNK data. However, it is apparent that there is a main effect of sex on withdrawal latencies on the hot plate at 56C. Given the minor sex differences noted above, it might be worthwhile maintaining your original figures with sexes identified.

7. PLOS authors have the option to publish the peer review history of their article (what does this mean?). If published, this will include your full peer review and any attached files.

Reviewer #1: No

Reviewer #2: No

---

## [Author Response · Author response to Decision Letter 1]

16 Mar 2024

Reviewer #1: Several studies by Jaclyn N. Highland and his team, including those with a comprehensive pharmacokinetics and pharmacodynamics of the HNKs are available suggesting the beneficial effect of HNK. Therefore, current contradictory finding is justifiable only if sufficient data has to be presented to rebut it mechanistically. If not, at least thermal nociception has to be assessed for HNK from three independent sources.

REPONSE: We based our pretreatment times on the bioavailability data provided in the Highland et al 2019 paper. Highland and colleagues found that HNK peaked at 15 mins and was still present in brain and plasma at 30 mins. HNK was undetectable after 1 hour. Therefore, we decided to use a 30 min pretreatment based on brain and plasma concentration. This is discussed on Page 16-17 lines 360-375 and Highland et al 2019 is also cited in the Drugs and Chemicals Section.

We understand the reviewer’s point that other studies have found antinociceptive effects of HNK in different assays. Most of these positive effects are found in chronic pain models rather than acute pain models, which is discussed in the introduction as well as the discussion. However, this is NOT the first time that independent research has failed to replicate or find positive effects of HNK – this was noted in our last response to reviewers. 

To compromise with Reviewer #1 we have decided to change the name of our manuscript slightly. The previous title was “(2R,6R)-hydroxynorketamine fails to produce antinociception in assays of acute pain-stimulated and pain-depressed behaviors in mice.” The new title is “Effects of (2R,6R)-hydroxynorketamine in assays of acute pain-stimulated and pain-depressed behaviors in mice”

Additionally, we added an “Other Considerations” paragraph in the discussion (pages 17-18, lines 378-398) to discuss the variables that could be considered confounding variables (this is in response to the reviewers last comment).

In the current study, authors have to revisit their assays of behavioral phenotyping and the quality of HNK. The quality certificate provided by authors is indicative of “quality at source” but not the “quality at bench”.

RESPONSE: We disagree with the Reviewer #1 on the statement that “the authors have to revisit their assays of behavioral phenotyping.” We have provided baseline tests with sex as a factor for ALL behavioral assays. Additionally, we have shown that the positive control morphine produced the appropriate antinociception response in all hot plate tests (figure 1). We also demonstrated that the positive control ketoprofen reversed acid-induce writhing (Figure 3) and pain-depressed locomotor activity (figure 4). Collectively, these data support that there are no issues with our behavioral assays or procedures. 

We have provided the certificate showing that HNK was pure. There is absolutely no reason to believe that the quality of HNK changed in our hands. We stored HNK per the manufacturer directions. Additionally, and most importantly, we had significant effects of HNK in the 56C hot plate at 30 mins (Fig 1C) and 24 hours (fig2) as well as a significant increase in rearing (fig 3F) during the acid-induced writhing assays (there was also a trend towards a reduction in writhing at 10 mg/kg; fig3E). Taken together, these data prove that HNK was active “at the bench”.

Please also check a technically robust paper on even the oral bioavailability of HNK in Mouse, rat, and dog . Highland, Jaclyn N., et al. "Mouse, rat, and dog bioavailability and mouse oral antidepressant efficacy of (2R, 6R)-hydroxynorketamine." Journal of Psychopharmacology 33.1 (2019): 12-24.

RESPONSE: We know this paper and based the pretreatment time of our studies on the bioavailability in mouse brain and plasma (we have added the citation to our drug section). Highland and colleagues found that HNK peaked at 15 mins and was still present in brain and plasma at 30 mins. HNK was undetectable after 1 hour. Therefore, we decided to use a 30 min pretreatment based on brain and plasma concentration. This is discussed on Page 16-17 lines 360-375.

Another study published in the FASEB journal suggest the Analgesic and Antinociceptive Effects of (2R,6R)-hydroxynorketamine (HNK) in Mice even after 24 hrs of drug administration(https://doi.org/10.1096/fasebj.2022.36.S1.R2181 )

RESPONSE: As stated in the previous revision, we collected most of these data prior to the Yost publication. It is important to note that Yost, only tested one acute assay (50C hot plate) – The other assays were for a chronic pain, which is very different. Yost et al found that HNK only produced an antinociceptive response at 24 hours (not at 10 mins, 1, 4, or 48 hrs). Based on the results of these studies, we added intermittent dosing studies that addressed the criticism of the 30 min pretreatment time as mice received a total of 4 HNK injections over 6 days. We have added a timeline of injections to figure 5 as there appears to be confusion on the 24 hour pretreatment time. We clearly tested HNK after 24 hours in 3 experiments (56C hot plate [fig 2], intermittent dosing 52C hot plate [fig 5] and intermittent dosing pain depressed behavior [fig 5]). We discuss the differences of the present paper and the Yost paper (mentioned above) at length on page 15 Lines 322-339.

Behavioral phenotypes of rodents may also be affected due to several technical, state and trait factors.

In the current study, both the male and female mice were used for assays. The presence of the opposite gender mice in the same housing, test arena and the odor/scent marks of opposite gender is sufficient to perturb the expected behavioral response. Therefore, it is suggested to describe the procedure employed to rule out such confounding factors. Such perturbations are more likely if neighboring mice are in estrous stage.

Following relevant paper of Nature Neuroscience is an example to explain that how important are the state and trait factors in pre-clinical behavioral pharmacology.

Georgiou, Polymnia, et al. "Experimenters’ sex modulates mouse behaviors and neural responses to ketamine via corticotropin releasing factor." Nature neuroscience 25.9 (2022): 1191-1200.

RESPONSE: All behavioral apparatus were cleaned with 50% EtOH to remove odor/scent marks. We have added a statement to the methods section for each assays. 

We have added a paragraph in the discussion (Page 17-18 Lines 378-398) to discuss the variables mentioned above.

Reviewer #2: Thank you for addressing the comments and including the additional data. Overall, the manuscript is a very interesting report and important for the field. There remain some edits that should be considered for clarity.

With regards to the statistical analysis, following a One -way ANOVA for this data set it is more appropriate to utilize a Dunnett multiple comparison test, with comparison made only to the control group 0 mg/kg rather than exploring differences between all of the doses used. Please make this change.

RESPONSE: We have now used Dunnett’s post hoc tests to compare the HNK writing data. This did not change any of the statistical results or the conclusions.

Figure 1, 3c - the legend indicating male has an error that needs to be corrected.

 RESPONSE: Thank for catching this issue. It has been corrected. 

It remains difficult to compare the ketoprofen data with that of HNK as figure 4e and 4f are presented differently to 4 c and 4d, I realize this is due to the dose response curve but it would make the point more salient if you could graph this as histograms as per 4c and 4 d

RESPONSE: We have made edits to the ketoprofen and HNK figures in figure 3 and 4 to make them as similar as possible for comparisons.

In addition, I'm not sure whether it is appropriate to separate the ketoprofen data by sex when you have decided to collapse the data across the sexes for the HNK data. However, it is apparent that there is a main effect of sex on withdrawal latencies on the hot plate at 56C. Given the minor sex differences noted above, it might be worthwhile maintaining your original figures with sexes identified.

RESPONSE: For the main manuscript, we have re-graphed the ketoprofen data to be consistent with the HNK data (see data has been moved to the supplemental material). We have added back in the sex data for the hot plate assay since there was an HNK mediated sex differences in th

---

## [Decision Letter · Decision Letter 2]

24 Mar 2024

Effects of (2R,6R)-hydroxynorketamine in assays of acute pain-stimulated and pain-depressed behaviors in mice

PONE-D-23-36070R2

Dear Dr. Porter,

We’re pleased to inform you that your manuscript has been judged scientifically suitable for publication and will be formally accepted for publication once it meets all outstanding technical requirements.

Kind regards,

Onder Tutsoy

Academic Editor

PLOS ONE

Additional Editor Comments (optional):

The revised paper has been re-assessed by two reviewers and the paper is ready for acceptance.

Reviewers' comments:

Reviewer's Responses to Questions

**Comments to the Author**

1. If the authors have adequately addressed your comments raised in a previous round of review and you feel that this manuscript is now acceptable for publication, you may indicate that here to bypass the “Comments to the Author” section, enter your conflict of interest statement in the “Confidential to Editor” section, and submit your "Accept" recommendation.

Reviewer #1: All comments have been addressed

Reviewer #2: All comments have been addressed

2. Is the manuscript technically sound, and do the data support the conclusions?

Reviewer #1: Yes

Reviewer #2: Yes

3. Has the statistical analysis been performed appropriately and rigorously? 

Reviewer #1: Yes

Reviewer #2: Yes

4. Have the authors made all data underlying the findings in their manuscript fully available?

Reviewer #1: Yes

Reviewer #2: Yes

5. Is the manuscript presented in an intelligible fashion and written in standard English?

Reviewer #1: Yes

Reviewer #2: Yes

6. Review Comments to the Author

Reviewer #1: New version of manuscript justify all the quarries and suggestions . The current version is acceptable for the publication.

Reviewer #2: It is my personal opinion that these studies were conducted in a rigorous and unbiased manner, and that the authors have considered many alternative avenues to derive the positive results that were predicted by the small number of articles currently available in the literature.

Thank you for your corrections. I have no further comments.

7. PLOS authors have the option to publish the peer review history of their article (what does this mean?). If published, this will include your full peer review and any attached files.

Reviewer #1: No

Reviewer #2: No

---

## [Editor Report · Acceptance letter]

8 Apr 2024

PONE-D-23-36070R2 

PLOS ONE

Dear Dr. Porter, 

I'm pleased to inform you that your manuscript has been deemed suitable for publication in PLOS ONE. Congratulations! Your manuscript is now being handed over to our production team.

Kind regards, 

on behalf of

Professor Onder Tutsoy 

Academic Editor

PLOS ONE